# Assessment of the Effects of Si Addition to a New TiMoZrTa System

**DOI:** 10.3390/ma14247610

**Published:** 2021-12-10

**Authors:** Mihaela-Claudia Spataru, Florina Daniela Cojocaru, Andrei Victor Sandu, Carmen Solcan, Ioana Alexandra Duceac, Madalina Simona Baltatu, Ionelia Voiculescu, Victor Geanta, Petrica Vizureanu

**Affiliations:** 1Public Health Departament, Faculty of Veterinary Medicine, “Ion Ionescu de la Brad” University of Life Sci-ences, 3 Mihail Sadoveanu Alley, 700490 Iasi, Romania; mspatarufmv@yahoo.com; 2Biomedical Sciences Department, Faculty of Medical Bioengineering, Grigore T. Popa University of Medicine and Pharmacy, 9-13 Kogalniceanu Street, 700454 Iasi, Romania; florinaivan90@yahoo.com (F.D.C.); duceac.ioana@icmpp.ro (I.A.D.); 3Department of Technologies and Equipments for Materials Processing, Faculty of Materials Science and Engineering, Gheorghe Asachi Technical University of Iaşi, Blvd. Mangeron, No. 51, 700050 Iasi, Romania; sav@tuiasi.ro; 4Romanian Inventors Forum, Str. Sf.P.Movila 3, 700089 Iasi, Romania; 5Preclinics Department, Faculty of Veterinary Medicine, “Ion Ionescu de la Brad” University of Life Sciences, 3 Mihail Sadoveanu Alley, 700490 Iasi, Romania; carmensolcan@yahoo.com; 6Quality Engineering and Industrial Technologies Department, Faculty of Industrial Engineering and Robotics, University Politehnica of Bucharest, 313 Spl Independentei, 060042 Bucharest, Romania; ioneliav@yahoo.co.uk; 7Engineering and Management of Metallic Material Processing Department, Faculty of Materials Science and Engineering, University Politehnica of Bucharest, 313 Spl Independentei, 060042 Bucharest, Romania; victorgeanta@yahoo.com

**Keywords:** titanium-based alloy, mechanical properties, biocompatibility

## Abstract

Ti-based alloys are widely used in medical applications. When implant devices are used to reconstruct disordered bone, prevent bone resorption and enhance good bone remodeling, the Young’s modulus of implants should be close to that of the bone. To satisfy this requirement, many titanium alloys with different biocompatible elements (Zr, Ta, Mo, Si etc.) interact well with adjacent bone tissues, promoting an adequate osseointegration. Four new different alloys were obtained and investigated regarding their microstructure, mechanical, chemical and biological behavior (in vitro and in vivo evaluation), as follows: Ti_20_Mo_7_Zr_15_Ta, Ti_20_Mo_7_Zr_15_Ta_0.5_Si, Ti_20_Mo_7_Zr_15_Ta_0.75_Si and Ti_20_Mo_7_Zr_15_TaSi. 60 days after implantation, both in control and experimental rabbits, at the level of implantation gap and into the periimplant area were found the mesenchymal stem cells which differentiate into osteoblasts, then osteocytes and osteoclasts which are involved in the new bone synthesis and remodeling, the periimplant fibrous capsule being continued by newly spongy bone tissue, showing a good osseointegration of alloys. A 3-(4,5-dimethylthiazol-2-yl)-2,5-diphenyltetrazolium bromide (MTT) assay confirmed the in vitro cytocompatibility of the prepared alloys.

## 1. Introduction

According to the European Society for Biomaterials, a biomaterial is “a material designed to interact with biological systems to evaluate, treat, enlarge or replace any tissue, organ or function of the body” [1]. Biomaterials include different materials designed or to be implanted in the body or to be used as surgical utensils. To contribute to the biomaterials field scientists work in close collaboration with medical doctors to find suitable, safe and effective materials for replacing some structure of the body [2]. The effects are obvious in medicine today: patients and doctors alike accept and expect long-term functional implants for treating almost any disease [3,4,5].

Although most biomaterials used in today’s patients are functional, some of them can cause side effects, hence the need to identify new materials that can actively interact with the body to achieve better results [6,7,8].

In human medicine, metals and their alloys together with other organic and inorganic materials have their origins a few thousand years ago, but only in the last century have they achieved important successes from a clinical point of view. The use of metallic materials in one field or another of the art depends on the relationship between the structure and the properties [9,10,11].

For the selection of biomaterials to make an implant, it is necessary to take into account economic, mechanical, electrical, environmental (chemical), safety (biological), thermal, surface, aesthetic, performance and research factors. The concept of biocompatibility refers to the interaction between the tissues and the physiological systems of the patient treated with a medical device. A biocompatibility assessment is part of the overall safety assessment of any device [12,13,14,15,16]. 

Titanium and titanium alloys are successfully used in medical applications, in dentistry, in the manufacture of fixed prostheses and skeletal prostheses, being considered a high quality implantable biocompatible material. Due to their low density, associated with good mechanical properties, titanium alloys are considered superior compared to other metallic alloys, having a strength/density ratio of about 300–400 MPa, which is higher than of that alloy steels, whose values vary between 150 and 350 MPa [9].

From a mechanical point of view, the conditions required for the implant metallic material are related to the capacity of good adaptation (fitting) during stress in conjunction with bone. Tensile strength is very important, but the elasticity must also be similar to that of bone. Otherwise, during tensile stress or bending the implant will act like a rigid fragment that will destroy the bone in which is implanted. For this reason, many studies have been done to improve the mechanical characteristics of biocompatible titanium alloys [10,11,12,13,14].

Researchers are concerned with choosing the right chemical composition for metallic alloys to achieve an elasticity modulus as low as possible (comparable with those of cortical bone which is about 7–30 GPa) [10,11,12], and for improving the tribological properties in the case of joints [13]. 

The research performed until now has revealed that plasticity, good tensile strength and low elasticity can be obtained using Mo, Nb, Ta, Sn, Pd, Hf and Zr as alloying elements in different percentages [14]. Due to the high chemical reactivity of titanium in air, methods to produce alloys have been developed, both by vacuum arc melting, mechanical alloying, cold isostatic pressing and sintering in vacuum [17] and using argon atmosphere for its protection during heat treatment. To achieve complete suppression of phase transformations, β stabilizer elements like Zr and Mo that can give a good combination of ductility, strength and strain-hardening rate have been added to titanium alloys [18]. The addition of over 11 wt.% Nb to a Ti-Mo alloy resulted in the reduction of the elastic modulus from 133 to 57 GPa [19]. The new series of Ti-Ta-Hf-Zr alloys displayed good behavior during plastic deformation, with the formation of multiple shear bands under compression (compressive yield strength of 1137 to 1158 MPa [20].

The addition of highly biocompatible alloying elements such as Mo, V, Zr, Ta to titanium contributes to improving the mechanical properties such as tensile strength and a gives a lower elasticity modulus compared to classical biocompatible materials [21,22,23,24].

Comparing the types of biomaterials (metals, ceramics and polymers), metals have important advantages as materials for implants considering their high strength and plasticity. Various metallic biomaterials have been tested and approved for biomedical applications, such as Ti and Ti-based alloys, Ni-Ti alloys or Co-Cr-Mo alloys, and applied in producing of bone plates, hip joints, cardiovascular stents, dental implants, or in improving the performance of surgical instruments [4]. Ti-based alloys have become very widely used for manufacturing orthopedic or dental devices under load-bearing applications because of their good biocompatibility and mechanical properties, low Young’s modulus and good corrosion resistance [3]. Some disadvantages of the Ti-based alloys are usually related to the materials’ wear properties. Comparing amorphous biomedical Ti-based alloys with crystalline Ti-based alloys higher strength and hardness was found in the amorphous materials and, as a consequence, a superior wear resistance under dry friction [25].

Since 1970s titanium and titanium alloys have been used in prosthetics because of their biocompatibility with organic tissues [26], determined by a low electrical conductivity. The values of conductivity influence the titanium electrochemical oxidation and determine the passivation of the implant by the generation of a thin oxide layer [27]. This layer of TiMoZrTaSi is mainly composed of TiO_2_/SiO_2_ oxides, with the SiO_2_ forming a stable film which prevents the corrosion of implants [28] while at the same time showing low reactivity with the macromolecules in the surrounding tissues [29]. Elias et al. [30] claimed a possible toxic effect of permanent implant applications resulting from the release of vanadium and aluminum compounds from the alloys. TiMoZrTaSi alloys with a higher proportion of Mo, Ta and Zr showed an increase of osteogenic activity in in vivo experiments, while at the same time having similar biomechanical characteristics with those of human bone tissue [30,31,32,33,34]. By adding Si to Ti alloys the physical strength and bonding strength of the metal alloy is improved [35] and the biocompatibility increased through directly bonding to bone tissue after implantation [10,11,26]. Kitsugi et al. [29] showed that a Ca-P layer (chemical apatite) is formed in direct contact with bone because of the absorption of phosphorus and calcium onto the surface of the formed passive layer, especially after a long period of implantation.

The paper presents some microstructural and biological aspects of a new Ti-Mo-Zr-Ta-Si alloy system. In order to enhance the mechanical and biological properties, biocompatible elements (Mo, Zr, Ta, Si) were added to a metallic matrix of pure titanium. The method used to obtain the alloys was melting in high vacuum equipment, using raw materials of high purity. In order to evaluate the alloys’ performances, microstructural analyses, mechanical testing and the testing of their chemical and biological characteristics (through in vitro and in vivo investigations) have been performed.

## 2. Materials and Methods

### 2.1. Material Preparation

For designing titanium alloys, both the properties of the alloying elements and their solubility in titanium must be taken into account. The chemical composition of the experimental alloys was chosen based on the literature [36,37] and on our own previous research [6,10,11,12,13,22]. It was found that by alloying titanium with Mo and Zr the value of the Young’s modulus decreased from 105 GPa (C.P. Titanium) to 69.02 GPa (for Ti_15_Mo_7_Zr_5_Ta) and to 51.93 GPa (for Ti_15_Mo_7_Zr_15_Ta) [22]. Adding silicon to Ti_15_Mo_0.5_Si alloy resulted in a decrease of the corresponding Young’s modulus to 19.82 GPa [22]. For higher amounts of silicon (1% wtSi) in Ti_15_MoSi alloy the Young’s modulus increased to 42.84 GPa. In the same time, for some of the studied alloys, the values of hardness have been reduced by 50% [10].

Based on these desired values of the modulus of elasticity, the biocompatibility of Ti_15_Mo_x_Si alloys (x = 0; 0.5; 0.75 and 1 wt.%) was then evaluated by in vitro and in vivo tests, the results indicating a successful osseointegration and no significant inflammatory reactions around implants in sheep at 62 days after implantation [11]. Similar biological studies have been conducted using Ti_20_Mo_x_Si alloys (x = 0; 0.5; 0.75 and 1 wt.%) and it was found that Ti_20_Mo_0.5_Si and Ti_20_Mo_0.75_Si alloys contribute to faster binding to the surrounding tissue, without any inflammatory or degenerative effects [10]. The next step for improving the mechanical characteristics of the titanium alloys was including in the chemical composition elements like Ta and Zr, which are known for their beneficial effects in Ti-based alloys and their biocompatibility [3,12,18,20].

The method chosen for obtaining titanium alloys is conditioned by the alloys’ behavior during the metallurgical process. When heated in air, titanium and its alloys will strongly interact with atmospheric gases, resulting in chemical combinations that form an intensely colored protective film, in direct relation with the temperature used. As the temperature increases, the oxidation rate intensifies, eventually producing a self-ignition effect. Titanium combines with nitrogen and hydrogen to form nitrides and hydrides, these effects being canceled through using a protective atmosphere during melting or heating. Depending on this behavior, titanium and titanium alloys used for medical applications can be successfully obtained, respecting the affinity restrictions for those gases, both at the laboratory and industrial level. The usual methods for obtaining titanium alloys are as follows: Electric arc melting furnace using non-fusible tungsten electrodes and a double wall water-cooled copper crucible, equipped with installations to ensure an inert protective atmosphere;Electron beam melting equipment, with a power supply of about 1000 kW, under high vacuum (10-4 torr), where the melted material solidifies in a water-cooled copper crucible;Vacuum arc melting equipment (VAR, under high vacuum (10-3–10-4 mbar) and in an argon-controlled atmosphere.

Considering these technological aspects, the experimental titanium alloys were prepared in an ABJ 900 vacuum arc remelting system (MRF, Allenstown, NH, USA) at the Laboratory for Obtaining and Refining of Metallic Materials, Faculty of Materials Science and Engineering, Politehnica University of Bucharest; (www.eramet.ro, accessed on 15 September 2021).

To analyze the influence of chemical composition on the material characteristics, four types of alloys were obtained, as follows: Ti_20_Mo_7_Zr_15_Ta, Ti_20_Mo_7_Zr_15_Ta_0.5_Si, Ti_20_Mo_7_Zr_15_Ta_0.75_Si and Ti_20_Mo_7_Zr_15_TaSi. To analyze the effect of silicon in titanium alloys, the content of the other alloying elements (molybdenum, zirconium and tantalum) was kept constant, then the Si content was increased from 0.5 to 0.75 and 1 wt.%, respectively. The titanium content value of the is calculated from the chemical balance equation.

To obtain the experimental Ti-Mo-Zr-Ta-Si system alloys, metallic raw materials with high purity (Ti—99.8%, Mo—99.7%, Zr—99.2%, Ta—99.5%, Si—99.5%) were mechanically processed and degreased for 20 min in an ultrasonic bath using an ethanol solution, then they were dried and handled with protective gloves for weighing and dosing according to the established recipe, based on the theoretical mass calculation. The raw material quantities used to make the TiMoZrTaSi system alloys are listed in Table 1, while the aspect of metal loads placed on the double-wall copper plate of the VAR installation and the mini-ingots obtained are shown in Figure 1 and Figure 2. The individual mass of the Ti-based alloys mini-ingots was about 50 g.

The raw materials loading on the copper plate of VAR equipment was done in increasing order of their specific density as follows: tantalum, molybdenum, zirconium and then silicon, titanium being last placed on these, to be first melted, and to form of a metallic bath that integrated the other chemical elements. 

During the melting operations, first a vacuum atmosphere of 4.5 × 10^−3^ mbar was applied, followed by purging the enclosure with inert gas (high purity Ar 5.3). This cycle was repeated four times to purify the working atmosphere in the VAR installation. For each mini-ingot, the melting process was repeated six times, on each side, in order to refine and homogenize the chemical composition. The technological parameters values established for obtaining the experimental Ti-based alloys were the following: melting power of min. 55 kVA; melting current of min. 650 A, 60% DS, three-phase voltage; vacuum level of 4.5 × 10^−3^ mbar; inert gas flow of 5 L/min [38].

The resulting chemical composition of the Ti-based alloys was measured by EDS analyses (Table 2), proving that the metallurgical process was correctly and sequentially performed, indicating a precise and homogeneous chemical composition. The density of the alloys was calculated and also listed in Table 1.

### 2.2. Microstructural Characterization Methods

Scanning electron microscopy (SEM-EDX Vega 2 LSH Tescan, Brno, Czech Republic) was used for compositional analysis. The scanning electron microscope is coupled with a QUANTAX-type EDX detector. The microscope is fully computer controlled and has an electron flux generated by a tungsten filament. The microscope reaches a resolution of 3 nm at 30 KV, with a magnification power between 13 and 1,000,000× in resolution mode with an acceleration voltage between 200 V at 30 kV, and a scan speed between 200 ns and 10 ms pixel^−1^. Metallographic analysis gives information on the micrographic structure of the alloy, its nature, shape, dimensions and distribution mode. An Zeiss Axio Imager A1 microscope (Carl-Zeiss-Strasse, Oberkochen, Germany) was used for high-precision optical image optical analyses.

### 2.3. Mechanical Properties

A CV-400 DM microdurimeter was used for analysis the aspects related to the microhardness of the alloys by the Vickers method, a widely used method in the biomaterials field.

An UMT 2 tribometer device (Bruker, Campbell, CA, USA) was used to determine the characteristics by indentation. This equipment has a Rockwell diamond penetrator (Bruker, Campbell, CA, USA), an angle to the indentor cone of 120 and a spherical tip of 200 μm radius that applies a force of 5 N. The Universal Microtribometer is used successfully and efficiently for the analysis of metallic materials (ferrous, non-ferrous, special, composite) or non-metallic (ceramic, plastic, wood, etc.), as well as special materials (different oils, lubricants or solid greases). By the method of identifying a trace of indentation the equipment records the movement of the diamond tip after it comes into contact with a surface and increases the value of the loading force until the tip penetrates the material. The equipment is also used in the analysis of thin and very thin layers or metallic and non-metallic coatings to determine their adhesion or mechanical properties.

### 2.4. In Vitro Cytocompatibility Assessment

In vitro biocompatibility tests on Ti-Mo-Zr-Ta-Si alloys were performed by incubation of the metallic materials with primary Albino rabbit fibroblasts and the cell metabolic activity was measured, using a 3-(4,5)-dimethylthiazol-2-yl)-2*H*-2,5-diphenyltetrazolium bromide (MTT) assay and comparing the results with a control to quantify the cytotoxic effects [39,40,41].

As medium for cells culture, standard Dulbecco’s Modified Eagle Medium (DMEM) including 4500 mg/mL glucose, 0.584 mg/L L-glutamine and 110 mg/L sodium pyruvate), bovine fetal serum (BFS, suitable for cell culture heat inactivated, sterilized and filtered), penicillin/streptomycin/neomycin (P/S/N, solution suitable for cell culture, formed by dissolution in sterile water of 5000 units penicillin, 5 mg streptomycin and 10 mg neomycin/mL, sterilized and filtered) were used as received from Sigma Aldrich (Merck KGaA, Darmstadt, Germany). Phosphate buffered saline solution (PBS solution, suitable for cell culture, sterilized by membrane filtration) and MTT were also purchased from Sigma-Aldrich.

The protocol used in our study includes the seeding of cells in a 12 well plate (with an initial density of cell at 5 × 10^3^ cells cm^−1^) in complete media, at 37 °C and 5% CO_2_, and growing up overnight. The samples of each material were decontaminated by immersion for 30 min in sterile 70% absolute ethyl alcohol solution. After that, the samples were successively washed with sterile water and solution for cell cultures (saline phosphate). The decontaminated samples were immersed in 2 mL of DMEM complete culture medium (DMEM with the addition of 10% BFS and 1% combination of penicillin, neomycin and streptomycin) and incubated for 24 h at 37 °C, atmospheric CO_2_ (5% concentration) and humidity of 97%. Finally, the sterilized sample were added to the pre-plated cells (in culture media) and maintained of for different periods of times (24 h, 48 h and 72 h). At each time of immersion, the MTT tests were performed and the cells viability was calculated.

At the end of incubation (24 h, 48 h and 72 h), the cells were treated with MTT as follows: the material and media from each culture well were removed and 0.5 mL of MTT working solution (0.25 mg/mL) in simple DMEM was added. The cells with MTT solution were incubated in dark condition for 2.5 h at 37 °C, humidity of 97%. After the period of incubation some reactions are produced: the tetrazolium ring is cleaved by the mitochondrial dehydrogenases of the viable cells, and produce MTT formazan crystals (purple, insoluble in aqueous solutions). To solubilize the crystals acidified isopropanol (1 mL/well) was added and a purple solution was obtained. The colored solution was measured at a wavelength of 570 nm using a Tecan plate reader adapted with Magellan V7.1SP1 software Sunrise model (Tecan, Männedorf, Switzerland). The absorbance results for culture cells incubated with metallic materials samples were normalizing to the negative control (culture cells incubated with PBS—same volume) and cell viability was calculated. All experiments were performed in triplicate, and the results are calculated as mean ± standard deviation (SD).

Cell viability was calculated as percentage, using the following Equation (1):Cell viability = (A_s_ × 100)/A_c_ [%](1)
where A_s_ is the absorbance registered for the cells cultured with metallic alloys and Ac is the control absorbance.

Additionally, the viable cell in the presence of metallic alloys was imaged after the staining with calcein-AM. The calcein-AM is a fluorescent dye which is able to color the live cells to fluorescence in green because of calcein-AM hydrolysis to calcein (by the acetoxymethyl ester from the intracellular esterases of the living cells). The cell-populated metallic alloys (two types of cells were used: Albino rabbit fibroblasts—primary cells, human MG63 osteosarcoma cell line) were cleaned by washing several times in HBSS and then were stained with fluorescent dye (40 min at 37 °C), as was recommended in the kit protocol. The hydrolyzed calcein (fluorescence) was imaged by using a DM5500Q fluorescence microscope (Leica, Heppenheim, Germany) equipped with a 455 nm filter for excitation and an emission of 530 nm.

### 2.5. In Vivo Biocompatibility Assessment

The experiments lasted a period of 60 days, using five rabbits of the species *Orychtolagus cuniculus* aged 8–10 months, both males and females, the control being implanted with a surgical orthopedic iron rod and the other four individuals with Ti_20_Mo_7_Zr_15_Ta, Ti_20_Mo_7_Zr_15_Ta_0.5_Si, Ti_20_Mo_7_Zr_15_Ta_0.75_Si and Ti_20_Mo_7_Zr_15_TaSi alloy 0.5 cm^2^ samples of quadrangular shape and a thickness of 0.2 mm. Before and after implantation, the rabbits were medically monitored and provided with conditions of comfort, feeding and preventive anti-infections treatment during 5 days, according to national and European legislation in the field of experimentation on animals [42]. The present study respects the current practices and legislation regarding animal protection and preclinical studies. The protocol was approved by the Ethical Committee of Iasi University of Life Sciences, registered under numbers 1492/03.12.2018 and no. 701/12.06.2020.

Domestic rabbits are rodents with rapid growth and easily maintained in individual or collective cages during experiments. Due to their size and docility, the rabbits are widely used in preclinical studies such as testing different alloys used in prosthetics, from studies focused on dental implants to post-implant infections control [29,30,42].

The surgery was conducted under general anesthesia using a xylazine-ketamine mixture, followed by trimming and disinfection of the cranial tibial region. An incision of approximately 0.7 cm was made and with the orthopedic electric drill, a bone breach (surgical canal) of approximately 5 mm in diameter and 6–8 mm deep was made, and each implant was inserted and fixed. The periosteum and regional fascia were then sutured, followed by the same on the skin (Figure 3).

Just after implantation and at the end of the experiment (60 days), X-ray exams were performed to establish the position of the implants and some aspects of the surrounding bone restoration. The radiographs were performed with an Intermedical Basic 4006 device (Intermedical S.R.L. IMD Group, Grassobbio, Italy) with voltage values between 48–52 kV being exposed at the intensity between 6.3–8 mAS. The images were recorded using a *X-CR* Smart Examion digital system (EXAMION GmbH, Berlin, Germany), saved in DICOM format and read using the Radiant program. 

At the end of experiment, CT evaluations were performed using a LightSpeed 16 device (General Electric, Milwaukee, Wisconsin, USA). The Hounsfield Unit Scale (UH) was used to identify the periimplant structures [43]. According to the CT scale, the air has the minimum possible density, minus 1000 UH, and appears black, the fat around minus 100, and the bone or calcifications, over +1000 UH. The metal structures of implants have a higher density than biological structures and produce artefacts.

Some slices of the implanted bones were taken from the euthanized animals and processed according to specific histological techniques, embedded in paraffin, sectioned with a microtome to a size of 0.5 μm and then stained with HE, HEA or subjected to special immunohistochemical staining.

For histological analysis parts of implanted bones were fixed in 10% formalin solution for 24 h and decalcified during 40 days with pH 7 15% EDTA. Slices (about 0.5 cm thick) were dehydrated with ethanol solutions of decreasing concentrations, then clarified in xylene and embedded in paraffin. After cutting with a microtome, 10 microscope slides from each paraffin block were selected being specific stained and observed under a CX41 microscope (Olympus, Hamburg, Germany).

Anti-osteopontin (GmbH Aachen Germany, antibodies, ABIN2774904), anti-MMP2 and anti–MMP9 (Santa Cruz, Biotechnology (C-20): SC-6840) were used to perform the immunohistochemical staining. After dewaxing the sections in xylene, hydrating in ethanol and microwaving for 10 min at 95 °C in 10 mmol pH 6 citric acid buffer, they were cooled for 20 min and after that twice washed in PBS for 5 min. The slices were treated with 3% hydrogen peroxide, and rinsed with PBS, after that being incubated overnight at 4 °C in humid atmosphere with primary antibodies, under dilution of 1:100. Next day, the slides were washed three times in PBS for 5 min before being incubated with the secondary antibodies. 

Goat anti-rabbit IgG secondary antibody was used to reveal the osteopontin activity of bone cells and for MMP2 and MMP9, a goat anti-mouse IgG secondary antibody was chosen. The microscope slides were developed in 3,3’-diaminobenzidine (DAB) being finally counterstained with hematoxylin.

## 3. Results and Discussion

### 3.1. Composition and Microstructure of Ti-Mo-Zr-Ta-Si Alloys 

Table 2 shows the elemental chemical composition of the Ti_20_Mo_7_Zr_15_Ta_x_Si (x = 0.5, 0.75, 1 wt.%) alloys as measured by energy dispersive x-ray (EDX) spectroscopy analysis. Table 2 presents an average of ten determinations, from five different areas of the samples (dimensions of 10 mm × 10 mm × 5 mm). The samples were homogeneous, and no inclusions were found.

Figure 4 shows the structure of the Ti_20_Mo_7_Zr_15_Ta_x_Si (x = 0.5, 0.75, 1 wt.%) alloys. The optical images show the alloy structures with β grain size and the α + β phase region. The beta phase was influenced by the addition of beta elements such as Mo, Ta and silicon. During cooling through the β phase field β grains appear, with slow cooling rates resulting in larger β grains.

The microstructure of the alloys is also always influenced by the elaboration method, with slow solidification producing thicker casting sections that exhibit larger β grains, thicker α plates, and larger α plate colonies, which are similarly aligned and have a common crystallographic orientation, respectively. Titanium-based alloys are predominantly affected the microstructure, depending on the amount of β- or α-stabilizing elements.

Diffractograms (Figure 5) of the Ti_20_Mo_7_Zr_15_Ta_x_Si (x= 0.5, 0.75, 1 wt.%), show the presence of the β phase in the obtained Ti-Mo-Zr-Ta-Si alloys, the alloys having a centered volume cubic structure.

### 3.2. Mechanical Properties of Ti-Mo-Zr-Ta-Si Alloys

Hardness measurements results of the Ti_20_Mo_7_Zr_15_Ta_x_Si (x = 0.5, 0.75, 1 wt.%) alloys were investigated and presented in Table 3 The measurements showed values between 305.34 HV and 274.64 HV. As the percentage of silicon increases, the hardness decreases.

The micro-indentation method is a common method for characterizing biomedical alloys. Table 4 shows the indentation results of the Ti_20_Mo_7_Zr_15_Ta_x_Si (x = 0.5, 0.75, 1 wt.%) alloys. The modulus of elasticity varied between 53.580 GPa (R5) and 63.882 GPa (R8). As the percentage of silicon increased, so did the modulus of elasticity. Comparing between Ti_20_Mo_7_Zr_15_TaSi (63 GPa) and classic titanium alloy C.P. (110 GPa), β-stabilizers such as Mo, Ta and Si significantly decreased the modulus of elasticity by almost 50%.

### 3.3. In Vitro Cytotoxicity of Ti-Mo-Zr-Ta-Si Alloys

Various Ti alloys have been tested in medical applications as orthopedic implant biomaterials or medical devices due to their excellent biocompatibility and interactions with human body, very good resistance to corrosion, fatigue strength, or their low Young’s modulus. In the last decades, new Ti alloys have been designed and tested with the aim of adding great corrosion properties and better mechanical characteristics to the known good biological compatibility properties [44,45].

In our study, a first test was performed in vitro, by incubating the obtained alloys with cell cultures (primary Albino rabbit fibroblasts were used for these experiments), with the aim of evaluating the metallic materials’ cytocompatibility. Cell cultures were incubated with the metallic alloys for 24 h, 48 h and 72 h and the cells’ viability was measured using a MTT assay. The data were reported with respect to the control and the calculated cell viabilities are presented in Figure 6.

The data presented in Figure 6 revealed that for all tested materials the calculated cell viability is more than 85%; according to ISO 10933-5 this indicates that all tested Ti-Mo-Zr-Ta-Si alloys are cytocompatible. Also, it can be observed that by including Si in the alloys the viability of the fibroblasts was not influenced greatly by this component. These data are in agreement with the results of other studies [46].

The samples of Ti-Mo-Zr-Ta-Si alloys were incubated with fibroblasts (2 × 10^4^ cells/ well) and after 72 h of culture (two types of cell types: Albino rabbit fibroblast primary cells, human MG63osteosarcoma cell line) the morphology was analyzed by optical microscopy in phase and fluorescence contrast. The resulting images are presented in Figure 7 and Figure 8.

Analyzing the fluorescence microscopy data it can be seen that the cells formed a monolayer in contact with the Ti-Mo-Zr-Ta-Si alloy samples (the materials are represented in the images by dark areas), and there are no significant differences compared with the growth of the control sample: the cells are elongated, with fibroblast morphology and a smaller and more spherical appearance for osteoblasts. Dark areas can be observed which are attributed to the location of the material; the material can induce some mechanical actions which determine the degradation of the cells’ monolayer integrity.

Regarding the cell growth density of the tested cells higher density is observed for line osteoblasts, compared to the primary fibroblasts; this can be explained by the behavior of the line cells (in our case MG63 osteoblasts) which grow faster than primary cells, which are more sensitive ones [47]. Analyzing the microscopy data for both types of cultures, it can be observed that the Ti-Mo-Zr-Ta-Si alloys do not influence the cell growth and morphology. By comparison to growth control, both fibroblast and osteoblast cells in culture look similar, indicating good cytocompatibility [48,49,50]. 

### 3.4. In Vivo Biocompatibility of Ti-Mo-Zr-Ta-Si Alloys

X-ray investigations of the implanted regions in experimental rabbits were pursued to establish the position of implants after 60 days of experiment and any possible reactions produced in the peri-implant tissues, but no abnormal radiological changes were identified, either in the control or the implanted rabbits (Figure 9).

Although the rabbits were of similar age and weight, the bone radiodensity highlighted by CT-scans differs from one individual to another. Using Hounsfield Units to measure the density of the tissues [51,52] an average of 1700 UH was obtained for compact bone tissue and between 300–931 UH for peri-implanted tissues in control and implanted rabbits.

Regarding the radiodensity of the periimplant-surrounding tissues after 60 experiments, in the control rabbits, the newly formed tissue in the implantation gap measures about 793 UH (compact bone is about 1500 UH) and the radio-opacity is between 633–931 UH (in compact bone it is about 1517–1712 UH) in the case of Ti_20_Mo_7_Zn_15_TaSi alloy and about 400–651 UH (compact bone is between 2274–2551 UH) for Ti_20_Mo_7_Zr_15_Ta alloy (Figure 10). A lesser periimplant radiopacity of up to 300–599 UH (and between 1718–1803 UH at the tibial compacta) was counted in the case of Ti_20_Mo7Zr_15_Ta_0.75_Si and about 305–620 UH (the tibial compacta is about 1291–1475 UH) for the Ti_20_Mo_7_Zr_15_Ta_0.5_Si alloy, respectively (Figure 10).

The radioopacity identified in the newly tissues around the implants was associated with fibrous intramembranous ossification tissues (in the case of Ti_20_Mo_7_Zr_15_Ta_0.75_Si and Ti_20_Mo_7_Zr_15_Ta_0.5_Si alloys) being strengthened by cartilage islands and ossification punctures in the case of Ti_20_Mo_7_Zr_15_Ta and Ti_20_Mo_7_Zr_15_Ta1Si alloys, as confirmed by histological analyses [10,11,53]. 

In the control rabbit, the peripheral area of the periosteum is characterized by a thickened cambium with mesenchymal stem cells, from which osteoblasts are formed, that synthesize the preosein and collagen fibers (Figure 11). Preosein is mineralized by osteoblast activity and produces ossein which structures the bone traveae, while the mineralized ossein delimitates lacunae into which the osteocytes locate (Figure 11, first row of images).

In the case of Ti_20_Mo_7_Zr_15_Ta alloy, in the implantation gap and periimplant area a layer of mesenchymal stem cells is observed (Figure 11, first row of images) followed by osteoblasts that are involved in bone synthesis (Figure 11, middle row of images). The periimplant fibrous capsule is continued by newly spongy bone tissue (Figure 11, last row of images).

In the inner layer of the fibrous capsule that surrounds the Ti_20_Mo_7_Zr_15_Ta_0.5_Si and Ti_20_Mo_7_Zr_15_Ta_0.75_Si alloys there are a large number of mesenchymal cells which are found in the bone areolae too (Figure 11, last row of images). The newly formed bone tissue contains a small number of bone lamellae, and the remodeling processes can be observed into the bone traveae (Figure 11, middle row of images). In the peri-implant area, the intramembranous centers of ossification occur, consisting of many proliferated mesenchymal stem cells that are found as osteoblasts in the new bone areolas, too (Figure 11, last row of images).

In the case of Ti_20_Mo_7_Zr_15_TaSi alloy, both intramembranous and endochondral types of ossification were observed in the peri-implant area. At the periphery of the regenerating bone, a layer of mesenchymal cells was found, continued by a succession of immature bone tissue cells with high cell density (Figure 11, first row of images), active osteoblasts, a mineralization line and mature bone tissue with lower cell density (Figure 11, middle row of images), plus mineralized matrix and osteocytes located in the bone lacunae. Osteoblasts show linearly arrangement at the periphery of the lamellar tissue (Figure 11, middle row of images).

In all cases, indirect regeneration (healing) processes occur in the peri-implant area consisting of endochondral and membranous ossification areas (Ti_20_Mo_7_Zr_15_TaSi), with the newly formed bone tissue being intensely vascularized (Figure 11, last row of images).

Osteopontin (OPN), secreted by macrophages, has binding properties to bone mineralized particles and bone wound margins created by drilling. It is thought that the interfacial OPN contributes to the adhesion of cells, is involved in cell signaling, and the mineralization of the intercellular matrix that is required for completely fusion of new bone with the pre-existing bone at drilling site edges [54].

Osteopontine (OPN) was highlighted in all experimental groups and in the control one. In the control group, the expression of OPN, MMP2 and MMP9 is moderate. An overexpression of OPN is observed in the bone interface area in the Ti_20_Mo_7_Zr_15_Ta_0.5_Si, Ti_20_Mo_7_Zr_15_Ta_0.75_Si and Ti_20_Mo_7_Zr_15_TaSi groups (Figure 12, first row of images). MMP2 and MMP9 recorded overexpression in all experimental groups (Figure 12, middle and last rows of images) During the processes of bone formation and its resorption the adhesion of some molecules (arginine-glycine-asparagine sequences) such as fibronectin, fibrinogen, vitronectin, type I collagen, OPN, or bone sialoprotein occurs, necessary to bind the osteoblasts or osteoclasts to the rebuilt surfaces [55].

The overexpression of OPN in cells represents a signal of osteogenesis in our experiments as well. OPN is produced by all types of bone cells, including differentiated osteoblasts and osteocytes [56]. OPN activity of fibroblast cells may be found in the case of embryonic stroma [57] or during the bone healing process [58]. Similarly, OPN is involved in osteoclast genesis and activity. OPN expression is regulated during the differentiation of monocytes in macrophages, a process that probably occurs when the monocytes migrate into tissues [59].

MMPs belong to the group of zinc-dependent endopeptidases which are the enzymes involved in the degradation and resorption of extracellular matrix compounds (Figure 12). The expression of both types of MMPs was also observed in our experiments (Figure 12, middle and last rows of images), in the periostal and periimplant cells. MMPs represent the significant regulators of the cellular and physiological processes of the body that are involved in processes like tissue repair, angiogenesis and morphogenesis. Similarly, they are expressed in the case of cancer, cardiovascular diseases and articular inflammatory or degenerative disorders or others [60,61,62]. For example, the expression of MMP-2 is observed during bone embryonic development, tissue recovery and carcinogenesis [63] while MMP-9 is known to be involved in bone remodeling based on the activity of osteoclasts [64,65].

## 4. Conclusions

For the design of new alloys based on the TiMoZrTaSi system, chemical elements that can provide properties specific to medical devices, such as biocompatibility, a low level of toxicity on the human body and mechanical characteristics similar to those of bone have been selected. Different batches of Ti_20_Mo_7_Zr_15_Ta_x_Si (x = 0.5, 0.75, 1%) alloys were prepared.

The experimental alloys were manufactured under the very good protection conditions provided by melted metal. Regarding the mechanical properties, the effect of adding the element silicon (a β-stabilizer) is very well highlighted in the Ti_20_Mo_7_Zr_15_Ta system as the percentage increases, the hardness decreases, and the modulus of elasticity increases slightly.

The cytocompatibility of the Ti-Mo-Zr-Ta-Si alloys was determined out by incubation of the metallic materials with fibroblasts and osteoblasts. Cell viability values and morphology analysis indicated no effect on cell growth, with both types of tested cells having characteristics comparable to their control cultures after incubation with Ti-Mo-Zr-Ta-Si alloys.

In all cases, the presence of compact lamellar bone and of mature osteocytes near the surface of the implants indicates a good biocompatibility and certainly, the presence of the implant didn’t interfere with new bone remodeling. The overexpression of OPN, MMP-2 and MMP-9 in the experimental group was registered both in the periosteum and periimplant area, the periimplant cells showing intense osteoblast and osteocyte activity during the new bone formation and the osteoclast genesis and osteoclast activity required for remodeling the architecture of the new bone.

## Figures and Tables

**Figure 1 materials-14-07610-f001:**
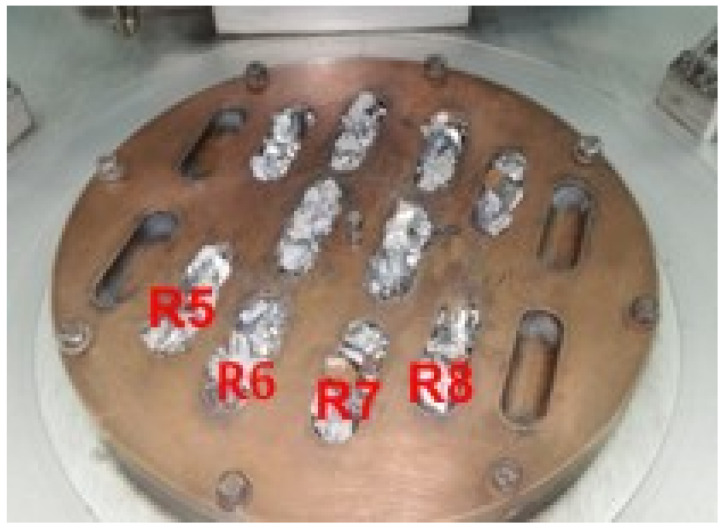
Metallic raw materials placed on the copper plate of VAR equipment.

**Figure 2 materials-14-07610-f002:**
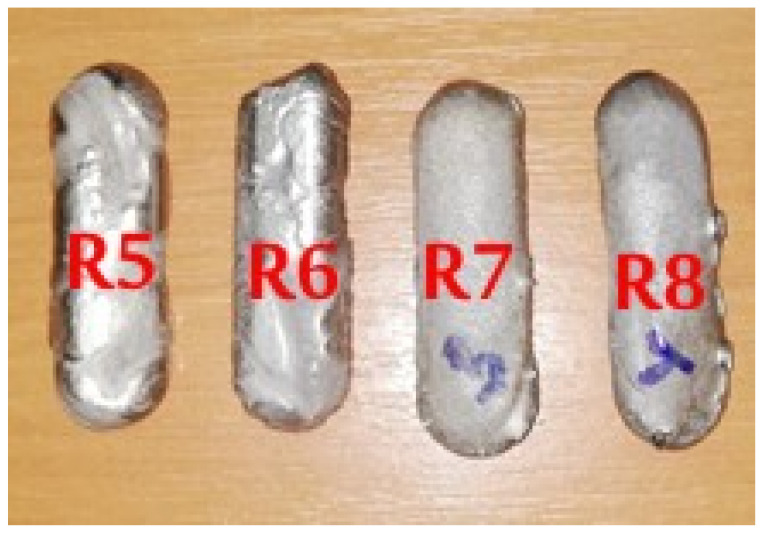
Mini-ingots of the alloys from TiMoZrTaSi system.

**Figure 3 materials-14-07610-f003:**
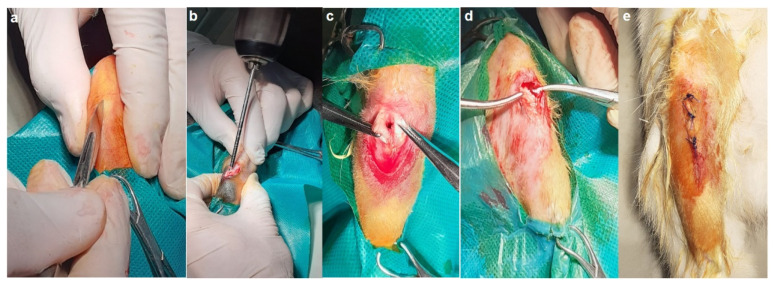
The surgical steps in alloys implantation: (**a**) sectioning of the skin and local fascia, (**b**) forming of implanting channel, (**c**) implant fixation, (**d**) suturing the periosteum, (**e**) fascia and the skin.

**Figure 4 materials-14-07610-f004:**
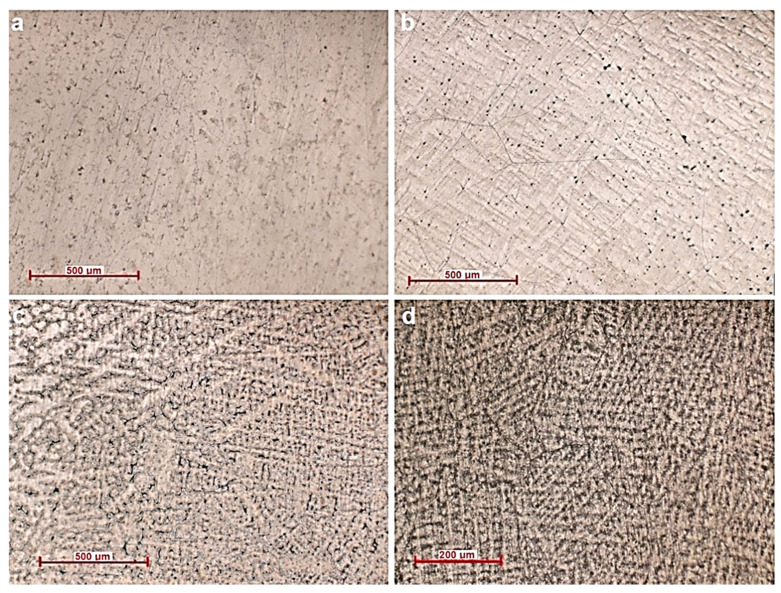
Optical microscopy images (**a**) Ti_20_Mo_7_Zr_15_Ta, (**b**) Ti_20_Mo_7_Zr_15_Ta_0.5_Si, (**c**) Ti_20_Mo_7_Zr_15_Ta_0.75_Si, (**d**) Ti_20_Mo_7_Zr_15_Ta_1_Si.

**Figure 5 materials-14-07610-f005:**
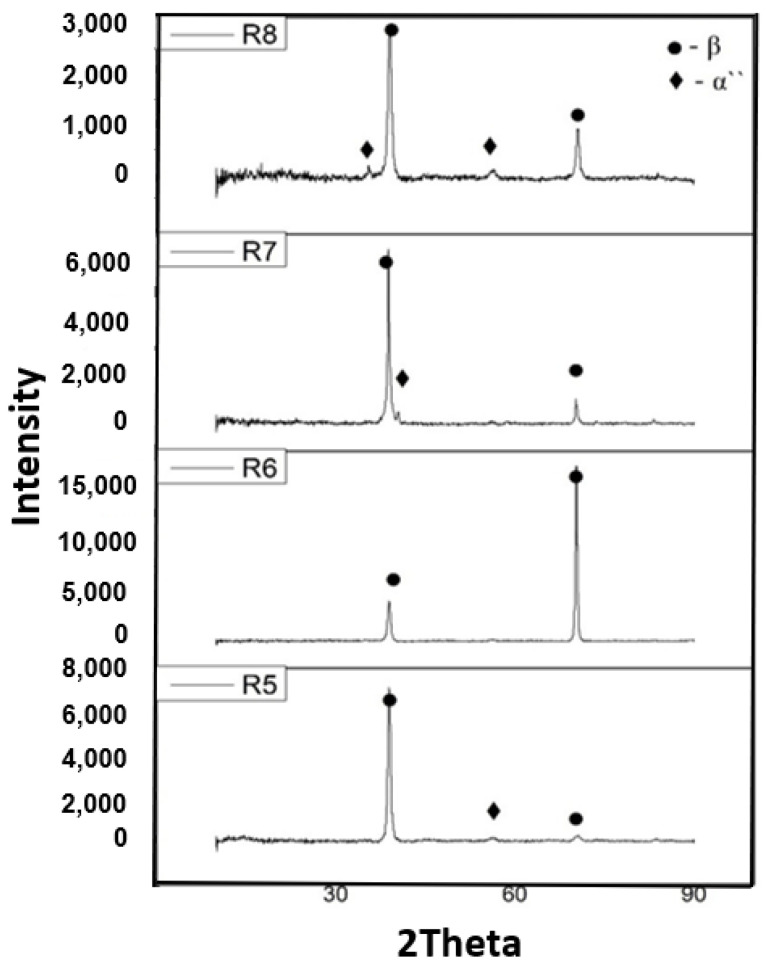
Diffractograms of experimental alloys: R5—Ti_20_Mo_7_Zr_15_Ta, R6—Ti_20_Mo_7_Zr_15_Ta_0.5_Si, R7—Ti_20_Mo_7_Zr_15_Ta_0.75_Si, R8—Ti_20_Mo_7_Zr_15_TaSi.

**Figure 6 materials-14-07610-f006:**
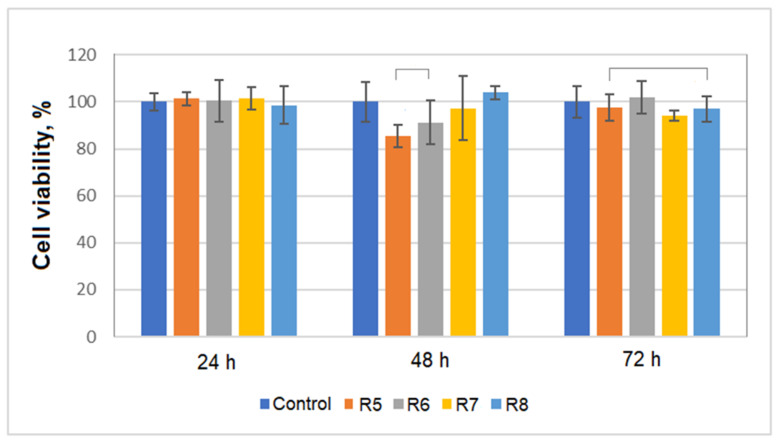
Mitochondrial activity measured via the MTT assay for albino rabbit fibroblast cultures exposed to metallic alloys for different times. Cell viability studies were carried out in triplicate (*n* = 3) for each experiment and analyzed by means of one-way ANOVA. A P-value of less than 0.01 was accepted as significant.

**Figure 7 materials-14-07610-f007:**
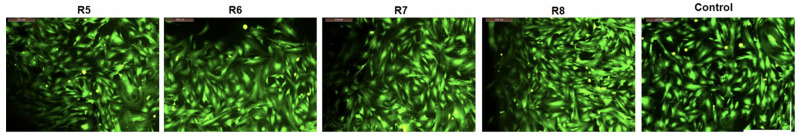
Phase and fluorescent microscopy (calcein AM) for materials at 72 h cell cultured with albino rabbit fibroblasts (10× objective magnification).

**Figure 8 materials-14-07610-f008:**
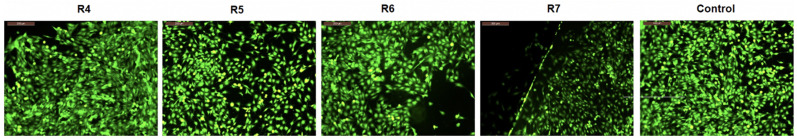
Fluorescent microscopy (calcein AM) results for Ti-Mo-Zr-Ta-Si alloys in contact with cell cultures for 72 h (human MG63, 10× objective magnification).

**Figure 9 materials-14-07610-f009:**
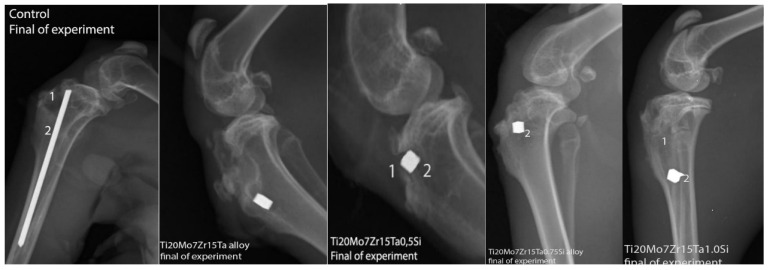
X-rays of control and experimental rabbits after 60 days of experiment, 1-alloy, 2-implantory breach.

**Figure 10 materials-14-07610-f010:**
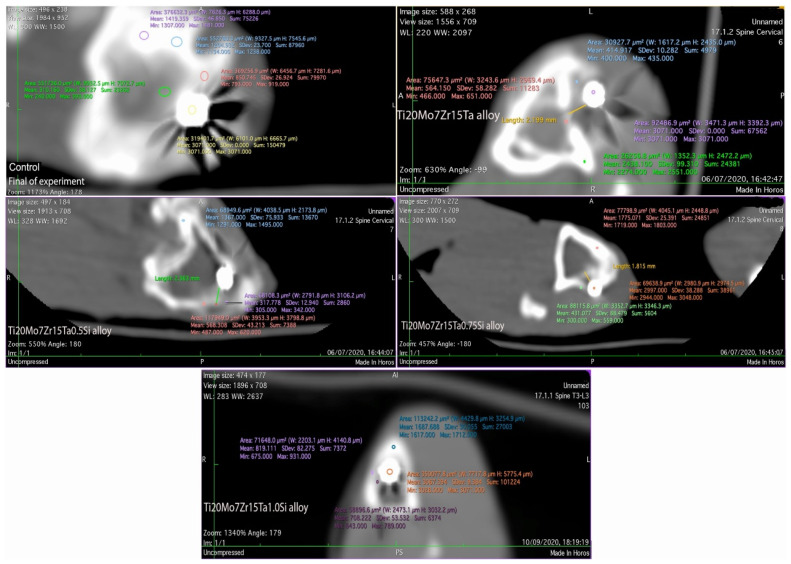
CT-scan of the implanted area in control and experimental rabbits after 60 days of experiment.

**Figure 11 materials-14-07610-f011:**
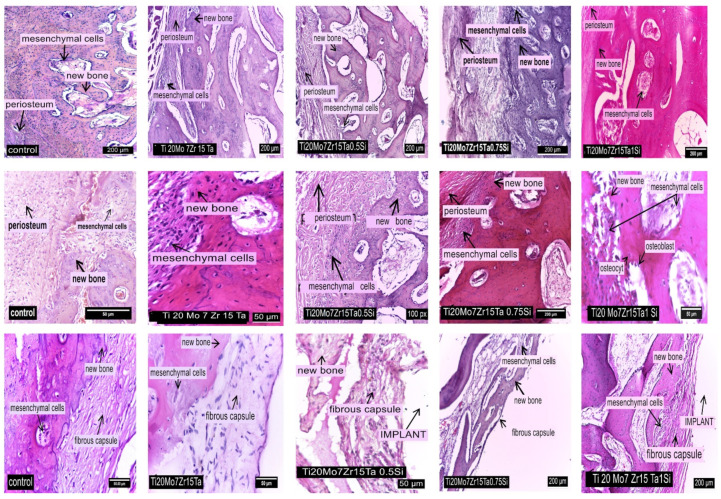
First row of images: The periosteum nearby the implant, in control and implanted rabbits; HE stain; Middle row of images: Some aspects concerning osteogenesis of new bone between periosteum and alloys in control and implanted rabbits, HE stain; Last row of images: The peri-implant aspects of the fibrous capsuleʼ structure and the new bone maturation in control and implanted rabbits, HE stain.

**Figure 12 materials-14-07610-f012:**
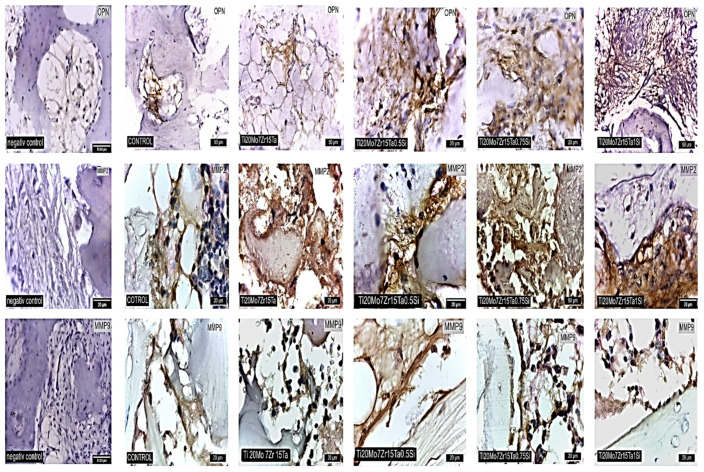
Stain for OPN and MMP2 and MMP9 in control and experimental rabbits nearby the alloy: First row of images: IHC stain, positive cells for osteopontin (OPN); Middle row of images: ICH stain, positive cells for metalloproteinases 2 (MMP2); Last row of images: ICH stain, positive cells for Metalloproteinases 9 (MMP9).

**Table 1 materials-14-07610-t001:** Data regarding the samples obtained in VAR process.

Sample	Designed Composition	Element Mass, g	Process Efficiency, %	Density g/cm^3^
R5	Ti_20_Mo_7_Zr_15_Ta	Ti = 31.5; Mo = 7.5; Zr = 3.5; Ta = 7.5	99.92	7.64
R6	Ti_20_Mo_7_Zr_15_Ta_0.5_Si	Ti = 31.25; Mo = 7.5; Zr = 3.5; Ta = 7.5; Si = 0.25	98.86	7.65
R7	Ti_20_Mo_7_Zr_15_Ta_0.75_Si	Ti = 31.13; Mo = 7.5; Zr = 3.5; Ta = 7.5; Si = 0.37	98.88	7.66
R8	Ti_20_Mo_7_Zr_15_TaSi	Ti = 31; Mo = 7.5; Zr = 3.5; Ta = 7.5; Si = 0.5	99.68	7.67

**Table 2 materials-14-07610-t002:** Chemical composition for Ti-Mo-Zr-Ta-Si alloys obtained.

Sample		R5	R6	R7	R8
Average chemical composition	Ti (wt.%)	58.35 ± 0.1	59.25 ± 0.2	57.86 ± 0.1	57.23 ± 0.1
	Mo (wt.%)	19.00 ± 0.2	18.50 ± 0.3	19.50 ± 0.1	19.83 ± 0.3
	Zr (wt.%)	8.15 ± 0.1	7.00 ± 0.1	6.85 ± 0.1	6.93 ± 0.1
	Ta (wt.%)	14.50 ± 0.3	14.80 ± 0.1	15.04 ± 0.3	14.98 ± 0.2
	Si (wt.%)	-	0.45 ± 0.1	0.75 ± 0.1	1.03 ± 0.1

**Table 3 materials-14-07610-t003:** The hardness values of Ti20Mo7Zr15TaxSi systems.

Alloys	Ti_20_Mo_7_Zr_15_Ta	Ti_20_Mo_7_Zr_15_Ta_0.5_Si	Ti_20_Mo_7_Zr_15_Ta_0.75_Si	Ti_20_Mo_7_Zr_15_TaSi
HV	305.34 ± 2.5	339.24 ± 2.2	315.27 ± 2.3	274.64 ± 2.9

Five determinations were carried out in different areas of each sample with dimensions of 10 mm × 10 mm × 5 mm.

**Table 4 materials-14-07610-t004:** Micro-indentation results.

Sample	Loading Deformation (N)	Release Deformation (μm)	Young Modulus (GPa)	Stiffness (N/μm)	Specimen Poisson Ration
R5	13.521 ± 0.1	6.759 ± 0.3	53.580 ± 0.3	4.531 ± 0.1	0.23 0.23 0.23 0.23
R6	13.530 ± 0.3	6.814 ± 0.2	54.256 ±0.4	4.641 ± 0.1	
R7	13.524 ± 0.2	5.992 ± 0.4	56.383 ± 0.1	4.269 ± 0.1	
R8	13.535 ± 0.3	6.736 ± 0.3	63.882 ± 0.2	5.643 ± 0.2	

Five determinations were carried out in different areas of each sample with dimensions of 10 mm × 10 mm × 5 mm.

## Data Availability

Data sharing is not applicable.

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
