# Peer review of "Assessment of the Effects of Si Addition to a New TiMoZrTa System"

_materials, 2021, doi:10.3390/ma14247610_

Round 1

Reviewer 1 Report

Please find hereby my review of the article untitled “Assessment of the Si addition on a new TiMoZrTa system.” by Mihaela-Claudia Spataru et al. in Materials

The submitted manuscript has a very interesting topic and a real scientific interest.

The paper presents some microstructural and biological aspects of a new Ti-Mo-Zr Ta-Si alloying system. In order to enhance the mechanical and biological properties, bio-compatible elements (Mo, Zr, Ta, Si) were added in the metallic matrix of pure titanium. 118 The obtaining method was melting in high vacuum equipment, using the high purity of raw materials. In order to evaluate the alloys performances, microstructure analyses, mechanical testing, and the testing of their chemical and biological characteristics (through in vitro and in vivo investigation)

The first comment concerns the form of the manuscript:

The manuscript needs to be edited carefully (typographical errors…).

You must show other data for the interpretation and the security in the biocompatibility chapter.

A s LDH or LDH+, DNA concentration, apoptosis, countering cells ….

You must show experiment for longer time as J10

Line 321 µM I presume? (and not cm?)

Figure 11

Improve the quality of colorations and the size of the image, difficult to compare

Figure 12

Improve the quality of ihc and the size of the image, difficult to compare

Your controls were largely stained? Its control antibodies? if not show them

Methods? no method for colorations and IHC?

The discussion about biology is quasi absent

Author Response

On behalf of the manuscript authors, I would like to sincerely thank you for your kind answer and valuable advices given to improve this research paper. 

Reviewer 2 Report

Spataru et al. evaluated Si-added TiMoZrTa alloys with extensive structural, mechanical, and in vitro & in vivo biological characterizations. The study was well designed. However, some key details about experiments, data analyses and presentation is missing/appears questionable. Specifically, the following points should be addressed prior to the acceptance of this manuscript for publication – 1. The use of English language needs major improvement. There are many grammatical errors and inappropriate uses of English. It is strongly recommended to have the manuscript proofread by a native speaker. 2. Fig 1 & 2 – resolution of the images is too low. 3. Table 1 – how was the process efficiency determined? Another key physical property – the alloys’ density – should be measured and compared for the different groups. 4. Figure 4 – it’s really difficult to see the grains with all the scratches on the sample surface. Further polishing following by etching would increase the visibility of grain boundaries. 5. Table 3 & 4 – how many measurements were made for each sample group? The authors should provide the average and standard deviation values for at least 3 measurements for each condition. 6. Figure 6 – what are S1-S4 in the figure? Same issue for the other figures for bio experiments. Was there any statistical analysis carried out? 7. Statistical analysis – there should be a subsection in Section 2 describing the statistical analysis methods employed and how statistical significance was indicated in figures. 8. Figure 7 & 8 – the length represented by the scale bar needs to be given. It seems that there were some dead cells in each image, and that the red fluorescence was not sufficiently bright and masked by the green color. Please verify. Also, it would be helpful to have quantitative data on the % of live cells. 9. Line 496 & 517 – contradicting and false statements on the source of OCN were made. Please verify and make corrections. 10. Conclusions – this section should be shortened by removing some of the descriptions to methods/discussion. It should only summarize the key findings and implications for bone repair applications.

Author Response

(The authors gave the same response as above.)

Round 2

Reviewer 1 Report

Now accept in this present form and the responses to reviewer

Reviewer 2 Report

the authors have addressed most of my points and the manuscript can be accepted for publication.